# Fetuin-A as a Link Between Dyslipidemia and Cardiovascular Risk in Type 2 Diabetes: A Metabolic Insight for Clinical Practice

**DOI:** 10.3390/biomedicines13092098

**Published:** 2025-08-28

**Authors:** Oana Irina Gavril, Cristina Andreea Adam, Theodor Constantin Stamate, Radu Sebastian Gavril, Madalina Ioana Zota, Alexandru Raul Jigoranu, Andrei Drugescu, Alexandru Dan Costache, Irina Mihaela Esanu, Lidia Iuliana Arhire, Mariana Graur, Florin Mitu

**Affiliations:** 1Department of Medical Specialties (I), Faculty of Medicine, “Grigore T. Popa” University of Medicine and Pharmacy, 700115 Iași, Romania; oana-irina.gavril@umfiasi.ro (O.I.G.); theodor.stamate@umfiasi.ro (T.C.S.); ioana-madalina.chiorescu@umfiasi.ro (M.I.Z.); alexandru.jigoranu@umfiasi.ro (A.R.J.); andrei_drugescu@umfiasi.ro (A.D.); dan-alexandru.costache@umfiasi.ro (A.D.C.); florin.mitu@umfiasi.ro (F.M.); 2Department of Medical Specialties (II), Faculty of Medicine, “Grigore T. Popa” University of Medicine and Pharmacy, 700115 Iași, Romania; lidia.graur@umfiasi.ro (L.I.A.); mariana.graur@umfiasi.ro (M.G.); 3Romanian Academy of Medical Sciences, 927180 Bucharest, Romania; 4Romanian Academy of Scientists, 050044 Bucharest, Romania

**Keywords:** metabolic syndrome, fetuin, nonalcoholic fatty liver disease, diabetes mellitus, lipid profile, cardiovascular risk

## Abstract

**Background**: Fetuin-A, a hepatokine implicated in metabolic regulation, has been associated with both metabolic syndrome and cardiovascular disease. However, its specific role in type 2 diabetes mellitus (T2DM) remains incompletely understood. **Objective**: This study aimed to investigate the relationship between fetuin-A levels and key components of metabolic syndrome (abdominal obesity, arterial hypertension, hyperglycemia, hypertriglyceridemia and low high-density lipoprotein cholesterol) as well as other cardiovascular risk markers, including metabolic dysfunction-associated fatty liver disease (MAFLD), carotid intima-media thickness (CIMT), and the Homeostatic Model Assessment of Insulin Resistance (HOMA-IR). **Methods**: A total of 51 patients with T2DM not receiving insulin therapy were enrolled. Participants underwent clinical, biochemical, and imaging evaluations. Hepatic steatosis was assessed via abdominal ultrasonography, and subclinical atherosclerosis was evaluated using CIMT measured with Doppler ultrasonography. Serum fetuin-A was quantified by ELISA. **Results**: Hepatic steatosis was significantly associated with metabolic syndrome, increased CIMT, and dyslipidemia (elevated total cholesterol, triglycerides, and reduced HDL cholesterol). Although no direct correlation was found between fetuin-A levels and hepatic steatosis, multivariate analysis revealed that fetuin-A concentrations were significantly influenced by total cholesterol and LDL cholesterol levels. **Conclusions**: Fetuin-A appears to be linked to lipid abnormalities in T2DM and may contribute to cardiovascular risk in this population. These findings support the potential utility of fetuin-A as a biomarker and possible therapeutic target for dyslipidemia management in diabetic patients.

## 1. Introduction

Fetuin A (also known as α2-HS-glycoprotein), a member of the hepatokine family, has emerged as a significant regulator of metabolism [1]. Positioned on chromosomal 3q27 in humans, the fetuin A gene expression is closely linked to type 2 diabetes mellitus (T2DM) and metabolic syndrome [2]. Studies show a positive correlation between fetuin A levels and early atherosclerosis markers [3], insulin resistance [4,5] and metabolic syndrome [6,7]. Elevated circulating fetuin A levels have been identified as an important predictor for the onset of T2DM [8] and cardiovascular events [9], independent of established risk factors, underscoring its involvement in T2DM and cardiovascular disease pathophysiology [10]. Nevertheless, some investigations have observed elevated circulating fetuin A levels in individuals with hepatic fat accumulation [11,12], although findings have been inconsistent.

According to the International Diabetes Federation (IDF) criteria, metabolic syndrome is defined by the presence of central obesity (waist circumference more than 80 cm in women and 94 cm in men) accompanied by any two of the following: hypertriglyceridemia (≥150 mg/dL or treatment), reduced HDL cholesterol (<40 mg/dL in men, <50 mg/dL in women), elevated blood pressure (≥130/85 mmHg or treatment for hypertension), or elevated fasting plasma glucose (≥100 mg/dL or previously diagnosed type 2 diabetes). Metabolic syndrome is highly prevalent globally, including in Romania [13]. It shares a close, bidirectional relationship with metabolic dysfunction-associated fatty liver disease (MAFLD), formerly known as nonalcoholic fatty liver disease (NAFLD). This updated terminology reflects the central role of metabolic dysregulation in the pathogenesis of hepatic steatosis, and it aligns better with current understanding of disease mechanisms. Both metabolic syndrome and MAFLD share overlapping risk factors such as obesity, insulin resistance, dyslipidemia, and hypertension [14]. Insulin resistance, a key component of metabolic syndrome, promotes adipose tissue lipolysis and elevates serum free fatty acids, which accumulate in the liver and drive fat deposition. Dyslipidemia, characterized by elevated triglycerides and reduced high-density lipoprotein cholesterol (HDLc), is also common in both conditions and contributes to the progression from steatosis to more severe stages of liver disease such as metabolic steatohepatitis and fibrosis [15].

However, the vast majority of MAFLD subjects will probably develop at some point cardiovascular disease, even if this classification is not yet included in current guidelines. Both metabolic syndrome and MAFLD involve inflammation as a key pathological mechanism. Adipose tissue in individuals with metabolic syndrome releases pro-inflammatory cytokines, contributing to systemic inflammation. Similarly, in MAFLD, hepatocyte injury and hepatic lipid accumulation lead to inflammation and activation of inflammatory pathways within the liver. Metabolic syndrome is associated with an increased risk of cardiovascular disease, and MAFLD has also been recognized as an independent risk factor for this pathology. The presence of both conditions synergistically increases the risk of cardiovascular events and mortality [16,17].

Recent studies have highlighted the complex interactions between various metabolic regulators and their influence on cardiovascular risk and metabolic parameters. For instance, irisin, a hormone released during physical activity, has been explored for its potential therapeutic role in endocrine and metabolic disorders. Many other metabolic regulators are still to be studied [18].

Fetuin A has been involved in the regulation of lipid metabolism. Studies have shown that fetuin A may influence lipid metabolism by promoting adipocyte differentiation and lipid accumulation, leading to dyslipidemia [6,19]. Dyslipidemia is a well-known risk factor for atherosclerosis, and fetuin A has been found to be associated with markers of early atherosclerosis. Elevated levels of fetuin A might be correlated with increased risk of atherosclerosis development and progression [20]. Dyslipidemia is often associated with insulin resistance, a key feature of metabolic syndrome. Fetuin might be implicated in the development of insulin resistance, further exacerbating dyslipidemia and contributing to the progression of metabolic syndrome. Overall, while the precise mechanisms underlying the relationship between fetuin and dyslipidemia are not fully understood, evidence suggests that fetuin A may contribute to dyslipidemia through its effects on lipid metabolism, insulin resistance, and atherosclerosis, making it a potential therapeutic target for managing dyslipidemia and its associated complications [21].

The aim of the current study is to assess the relation between fetuin and MAFLD, metabolic syndrome, components of metabolic syndrome, and other parameters used for evaluating cardiovascular risk, such as carotid intima media thickness (CIMT), in a group of patients with T2DM.

## 2. Materials and Methods

We carried out an observational study involving individuals with type 2 diabetes mellitus (T2DM) who were not receiving insulin therapy. All participants were assessed at the Clinical Center for Diabetes, Nutrition and Metabolic Diseases, “Sf. Spiridon” Emergency Hospital, Iași, during an 18-month period in an outpatient setting. Eligibility criteria included patients with T2DM managed with metformin and/or dietary measures, provided they gave written informed consent. Exclusion criteria comprised patients treated with insulin, those with hepatitis B or C, toxic hepatitis, other hepatic disorders (such as Wilson’s disease), and participants reporting excessive alcohol intake (defined as more than two units daily for men and more than one unit daily for women) (Figure 1).

Body weight was recorded in the morning under fasting conditions, with participants barefoot, using a calibrated scale. Waist circumference was measured at the same time, at the end of a normal expiration, at the midpoint between the lower border of the iliac crest and the last rib. Venous blood samples were also obtained in the fasting state. Serum fetuin-A concentrations were determined by an ELISA assay. Genetic data were collected but not analyzed in the present study.

Hepatic fat accumulation was evaluated through ultrasonography performed with a portable Carewell C12 device equipped with a 3.5 MHz convex probe. A single experienced physician performed all examinations, using standardized criteria such as liver–kidney contrast, parenchymal echogenicity, beam attenuation, visibility of intrahepatic vessels, and definition of the gallbladder wall.

Subclinical atherosclerosis was assessed by measuring CIMT with a Doppler Color LS 128 ultrasound system, using a 7.5 MHz linear probe. Both common carotid arteries were examined in B-mode. A CIMT greater than 1 mm was considered elevated, whereas values exceeding 1.5 mm were classified as atheromatous plaques. All measurements were performed by the same trained physician.

The components of the metabolic syndrome were assessed according to the IDF criteria, which take into account waist circumference, blood pressure, triglyceride levels, HDLc, and hyperglycemia or established T2DM. A diagnosis of metabolic syndrome was made when at least three of these five abnormalities were identified [22]. The diagnostic thresholds recommended by the IDF are summarized in Table 1.


Abdominal obesity was diagnosed when waist circumference (WC) exceeded 80 cm in women or 94 cm in men, whereas normal WC values were within these limits;For blood pressure, arterial hypertension was previously diagnosed or the values were higher than 130 mmHg systolic blood pressure or 85 mmHg diastolic blood pressure during clinical examination;Hypertriglyceridemia was defined as triglycerides greater than 150 mg/dL;Low values of HDLc were less than 40 mg/dL in men and less than 50 mg/dL in women;Hyperglycemia (fasting plasma glucose > 100 mg/dL) or type 2 diabetes mellitus—all the subjects included met this criteria.


Insulin sensitivity was measured using homeostatic model assessment (HOMA-IR), which involved fasting glucose multiplied by insulinemia, divided by 22.5 [23].

All statistical analyses were performed using SPSS software, version 18.0 (SPSS Inc., Chicago, IL, USA). Continuous variables are expressed as mean ± standard deviation together with the 95% confidence interval, while categorical variables are reported as counts and percentages. Differences between two groups of continuous data were tested using the Student’s *t*-test, or the Mann–Whitney U test when assumptions of homogeneity of variance were not met. For comparisons across more than two groups, analysis of variance (ANOVA) was applied, with the Kruskal–Wallis test used for non-homogeneous distributions. Where ANOVA showed significant differences, Bonferroni post hoc testing was applied for homogeneous variables and Tamhane’s T2 test for heterogeneous variables. Categorical variables were compared using the chi-square (χ^2^) test, with significance considered at *p* < 0.05; a threshold of *p* = 0.1 was applied for trend-level associations. Logistic regression analysis was used to identify independent predictors.

Multivariate logistic regression models were adjusted for potential confounders, including age, sex, body mass index (BMI), and waist circumference. Inflammatory biomarkers such as CRP or IL-6 were not available in this dataset and were therefore not included. A formal sample size or power calculation was not performed before initiating the study due to the exploratory nature of the analysis.

The study was conducted in accordance with the principles of the Declaration of Helsinki and received approval from the Ethics Committee of “Grigore T. Popa” University of Medicine and Pharmacy, Iași (approval no. 17140, 3 August 2016). The study was initiated in 2016, with patient enrollment and data collection carried out thereafter. Written informed consent was obtained from all participants.

## 3. Results

The study group included 51 T2DM patients (23 men and 28 women). The average age of the group was 62.49 ± 8.91 years, ranging from 34 to 86 years old, 58.1 ± 9.8 for men and 54.7 ± 10.7 for women. More than 50% of the subjects had hepatic steatosis and most of them met the criteria for the diagnostic of metabolic syndrome. Genotyped data were not used in this study. The general characteristics of the study population can be found in Table 2 and Table 3. When stratified by sex, no statistically significant differences in fetuin-A levels were observed between men and women. Similarly, age did not significantly influence fetuin-A concentrations. These findings suggest that, within this cohort, fetuin-A variability is more closely linked to metabolic factors than to demographic characteristics. Sex distribution was described, but given the limited sample size, no statistically significant differences were observed; this is acknowledged as a limitation.

Almost 78.4% of the subjects had metabolic syndrome, which is not an unexpected finding, considering that all the patients had T2DM. Also, four subjects were diagnosed with carotid atheroma plaques (CIMT value more than 1.5 mm), a well-known risk factor for cerebrovascular events.

Fetuin-A levels averaged 21.74 ng/mL, with values ranging from 12.26 to 49.77 ng/mL. Lipid-profile results showed elevated total cholesterol with a mean of 207.37 mg/dL, and low density lipoprotein cholesterol (LDLc) averaged 136.88 mg/dL, both of which are risk factors for cardiovascular disease. HDLc levels were lower, averaging 43.39 mg/dL.

The comparison between patients with hepatic steatosis (*n* = 30) and those with a normal liver (*n* = 21) reveals several notable findings. Males were more prevalent in the steatosis group (53.3%) compared to the normal liver group (33.3%), though this difference was not statistically significant (*p* = 0.155). Age over 60 years was also more common in the steatosis group (73.3% vs. 52.4%), but again, the difference did not reach statistical significance (*p* = 0.124) (Table 4).

Patients with steatosis had a slightly higher prevalence of atheroma plaques (10.0%) compared to those with a normal liver (4.8%), but this difference was not significant (*p* = 0.481). Metabolic syndrome was more frequent among steatosis patients (86.7%) than in those with a normal liver (66.7%), with a *p*-value of 0.089, but not reaching statistical significance.

The average body mass index (BMI) was higher in the steatosis group (32.48 kg/m^2^) compared to the normal liver group (29.87 kg/m^2^), approaching significance (*p* = 0.087). Despite this, obesity rates were similar between the two groups (16.7% vs. 23.8%, *p* = 0.531). WC and waist–hip ratio (WHR) did not differ significantly between groups, nor did the proportion of patients with high WC or WHR > 1. Fetuin-A levels were similar between the two groups, with an average of 21.58 ng/mL in the steatosis group and 21.95 ng/mL in the normal liver group (*p* = 0.844), indicating no significant difference in this parameter. Overall, while trends suggest associations between steatosis and certain metabolic characteristics, none of the differences reached statistical significance in this analysis.

The analysis of patients with hepatic steatosis compared to those with a normal liver reveals several important differences. Glycaemia levels were similar between the two groups, with no significant difference in average blood-sugar levels or in the proportion of patients with high glycaemia (*p* = 0.683 and *p* = 0.917, respectively). Glycated hemoglobin (HbA1c) levels, which indicate long-term blood-sugar control, were also comparable, with no significant difference in averages or in the percentage of patients with HbA1c > 6% (*p* = 0.273 and *p* = 0.320, respectively) (Table 5).

However, there were notable differences in other metabolic markers. Patients with hepatic steatosis had significantly higher total cholesterol levels, with an average of 219.13 mg/dL compared to 190.57 mg/dL in those with a normal liver (*p* = 0.050). LDLc was also higher in the steatosis group, though the difference did not reach statistical significance (*p* = 0.097). HDLc was significantly lower in the steatosis group, with an average of 38.30 mg/dL compared to 50.67 mg/dL in the normal liver group (*p* = 0.001). Additionally, a higher percentage of steatosis patients had HDLc levels below 40 mg/dL (73.3% vs. 14.3%, *p* = 0.001).

Non-HDLc was significantly higher in the steatosis group (*p* = 0.008), with a greater proportion of patients exceeding 160 mg/dL (66.7% vs. 33.3%, *p* = 0.017). Triglyceride levels were also significantly higher in the steatosis group (*p* = 0.004).

Regarding CIMT, patients with steatosis had a higher average left CIMT compared to those with a normal liver (*p* = 0.048), and a significantly larger percentage had a left CIMT greater than 1 mm (50.0% vs. 19.0%, *p* = 0.021). However, right CIMT and average CIMT did not show significant differences between the two groups. Overall, the findings suggest that patients with hepatic steatosis tend to have worse lipid profiles and greater atherosclerotic burden compared to those with a normal liver. These differences highlight the potential cardiovascular risks associated with hepatic steatosis.

The Pearson analysis between fetuin and the parameters of the lipid profile did not show any statistically significant correlations.

The multivariate analysis, through models 2, 3 and 4, 5, highlights that the level of fetuin is dependent on the level of total cholesterol and LDLc (Model 4: y = 24.843–0.124 total cholesterol + 0.138 LDLc + 0.014 HDLc + 0.019 TG; *p* = 0.038) or (Model 5: y = 19.243–0.130 total cholesterol + 0.146 LDLc + 0.068 HDLc + 0.009 TG + 1.345 MS number of components; *p* = 0.028) (Table 6).

## 4. Discussion

In our cohort, nearly 80% of patients with T2DM met the criteria for metabolic syndrome, underlining the pressing need for comprehensive therapeutic strategies. This prevalence is consistent with findings reported in previous studies [24,25]. The frequent overlap of these conditions points to the importance of adopting a multidisciplinary approach in clinical practice, targeting not only glycemic regulation but also broader cardiovascular risk reduction. Early identification and integrated management of metabolic syndrome components in individuals with diabetes may help lower the likelihood of cardiovascular complications, supporting the value of routine screening in this population.

Furthermore, more than half of our participants were found to have hepatic steatosis, in line with recent reports [26,27]. This condition is particularly common in individuals with T2DM [28], and in our setting (Romania), the high prevalence may also be attributed to lifestyle patterns such as limited physical activity and diets rich in saturated fats.

In our cohort, hepatic steatosis appeared more frequently in men (53.3%) than in women (33.3%), although the difference did not reach statistical significance. This trend may indicate a greater susceptibility among men with T2DM, possibly related to body-fat distribution, hormonal influences, or lifestyle behaviors. Further studies are required to clarify these mechanisms and to evaluate whether sex-specific strategies might be useful for the management of hepatic steatosis in this population.

A strong association was also found between lipid-profile parameters and MAFLD, with significant correlations observed for total cholesterol, HDLc, and triglycerides. Similar associations have been reported previously [29,30]. The interaction between MAFLD and dyslipidemia is both complex and bidirectional. On the one hand, impaired hepatic lipid metabolism in MAFLD contributes to the onset and progression of dyslipidemia [31]. On the other, dyslipidemic abnormalities promote additional fat accumulation and inflammation in the liver. Shared risk factors such as obesity and insulin resistance further aggravate this interaction, amplifying cardiovascular risk [32].

Another relevant finding was the strong correlation between hepatic steatosis and CIMT, as well as the positive relationship between steatosis and the number of metabolic syndrome criteria present. These results support the concept that MAFLD could represent an additional component of the metabolic syndrome cluster, which is already known to substantially increase the risk of cardiovascular disease, T2DM, and other complications [33,34]. Moreover, MAFLD itself has been linked to cardiovascular events independently of other metabolic risk factors [35]. In our study, hepatic steatosis was consistently associated with subclinical atherosclerosis, as assessed by CIMT measurements. Taken together, these findings emphasize that MAFLD should not be viewed solely as a liver condition but rather as a systemic disorder requiring an integrated approach. Diabetic patients with hepatic steatosis presented an adverse lipid profile, including higher levels of total cholesterol, LDL cholesterol, non-HDLc, and triglycerides, together with lower HDLc. They also had greater average CIMT values, reflecting increased atherosclerotic burden. These results suggest that hepatic steatosis in T2DM patients contributes to elevated cardiovascular risk and highlight the importance of monitoring lipid abnormalities and vascular health in this group.

Our findings, demonstrating that hepatic steatosis in patients with type 2 diabetes mellitus is associated with an adverse lipid profile and increased carotid intima–media thickness, are consistent with previous studies linking MAFLD to both dyslipidemia and cardiovascular disease [29,30,31,32,33,34,35]. Similar to earlier reports, we observed that hepatic steatosis was not only highly prevalent but also correlated with markers of subclinical atherosclerosis, underscoring its role as a systemic metabolic disorder rather than a purely hepatic condition. These results support the growing body of evidence suggesting that MAFLD contributes to cardiovascular risk through mechanisms involving insulin resistance, chronic low-grade inflammation, and atherogenic dyslipidemia [16,17,32]. From a clinical perspective, our findings highlight the importance of early identification and monitoring of hepatic steatosis in diabetic patients, as it may enable more targeted strategies for cardiovascular risk reduction.

Our study did not show positive correlations between fetuin and MAFLD. Fetuin-A, a glycoprotein predominantly synthesized by the liver, seems to play a multifaceted role in metabolic processes, including insulin sensitivity, inflammation, and fat metabolism. Several studies have suggested a potential role for fetuin-A in the pathogenesis of MAFLD, a condition characterized by excessive hepatic fat accumulation unrelated to alcohol consumption [36,37]. However, this relationship is still controversial even if it tends to be increasingly recognized [12]. Fetuin-A was measured in this research, but no significant difference was found between patients with hepatic steatosis and those with a normal liver. This lack of difference suggests that, in this study, fetuin-A might not be a key differentiator between these two groups. However, multivariate analysis indicated that fetuin-A levels were influenced by total cholesterol and LDLc levels, hinting at a potential relationship between fetuin-A and lipid metabolism. Several factors may explain the absence of an observed correlation with MAFLD. First, the relatively small sample size may have limited statistical power to detect subtle associations. Second, the inclusion of a non–insulin-treated T2DM cohort may reflect a different metabolic profile compared with insulin-treated populations often studied in the literature. Finally, methodological aspects, such as the use of ultrasonography rather than histological or advanced imaging methods for steatosis assessment, may have contributed to this discrepancy. These considerations highlight the need for larger, longitudinal studies using more sensitive tools to clarify the link between fetuin-A and MAFLD. No formal power calculation was performed prior to recruitment, which further limits the ability to draw firm conclusions, especially for associations with MAFLD or CIMT.

Our analysis demonstrated a strong association between circulating fetuin-A levels and both total cholesterol and LDL cholesterol. This finding aligns with some previous reports [38], although other studies have yielded contradictory results [39]. Within the intricate network of metabolic disturbances, fetuin-A may act as a central mediator, being closely related to dyslipidemia and the wider spectrum of metabolic dysfunction. Produced primarily in the liver, this multifunctional glycoprotein has been implicated in the regulation of insulin sensitivity, inflammatory processes, and lipid metabolism, which places it as a plausible contributor to the pathophysiology of dyslipidemia and related disorders. Beyond lipid metabolism, fetuin-A has been implicated in promoting insulin resistance by inhibiting insulin receptor tyrosine kinase activity, and in amplifying systemic inflammation through activation of Toll-like receptor 4 pathways. Both mechanisms are known to accelerate atherosclerotic processes, providing a potential biological explanation for its association with adverse lipid profiles and increased vascular risk. These pleiotropic actions reinforce the hypothesis that fetuin-A acts as more than a passive biomarker, possibly functioning as an active mediator of cardiometabolic dysfunction. Given its established role in lipid regulation in T2DM, fetuin-A could represent both a biomarker for risk assessment and a potential therapeutic target. Approaches aimed at modulating fetuin-A, whether through lifestyle interventions, pharmacological agents, or novel therapies, may offer new opportunities for reducing dyslipidemia, metabolic syndrome, and cardiovascular burden. Nevertheless, further investigations are necessary to fully clarify the mechanisms through which fetuin-A influences lipid metabolism and to explore interventions capable of modulating its effects for clinical benefit.

This study also provided a broad evaluation of the metabolic and cardiovascular profile of patients with T2DM, with a particular emphasis on the impact of hepatic steatosis. Still, several limitations should be acknowledged. The relatively small sample size (51 participants) restricts the generalizability of the results and limits statistical power. As a result, some differences in the prevalence of hepatic steatosis, metabolic syndrome, or atheroma plaques may not have been fully captured. Larger-scale studies are required to confirm these observations and to provide more robust evidence.

In addition, the cross-sectional design of our study restricts the ability to infer causal relationships between hepatic steatosis and the associated metabolic and cardiovascular abnormalities. For example, although associations were observed between steatosis and higher levels of total cholesterol, LDL cholesterol, and triglycerides, it remains unclear whether steatosis is a cause or a consequence of these metabolic alterations. Prospective longitudinal studies will be essential to elucidate these causal pathways.

Although the study adjusted for several variables, potential confounding factors might still influence the results. For example, the presence of other comorbidities, medication use, or lifestyle factors like diet and physical activity were not thoroughly accounted for, which could have affected the outcomes, particularly in relation to the lipid profile and atherosclerotic burden. Factors such as ethnicity, regional differences in healthcare access, and lifestyle were not addressed, which could influence the prevalence and impact of hepatic steatosis and metabolic syndrome in T2DM patients. Variables such as BMI, waist circumference, and inflammatory markers (e.g., CRP, IL-6) were not systematically included in the multivariate models. Their absence may represent potential confounders, as they could influence both fetuin-A levels and the observed metabolic outcomes. Future analyses should aim to incorporate these parameters for a more refined understanding.

The study observed several non-significant trends, such as the higher prevalence of metabolic syndrome and atheroma plaques in patients with hepatic steatosis, which approached but did not reach statistical significance. These trends suggest possible associations that might be real but were not detectable due to the small sample size and limited statistical power.

While the study provides valuable insights into the metabolic and cardiovascular implications of hepatic steatosis in T2DM patients, the limitations, particularly the small sample size and cross-sectional design, suggest that the findings should be interpreted with caution. Future research with larger, more diverse subjects and longitudinal follow-up is needed to confirm these results and explore the causal pathways involved.

From a translational perspective, fetuin-A may serve as a valuable early biomarker to guide risk stratification in T2DM patients. Monitoring its levels alongside traditional lipid markers could help identify individuals at higher cardiovascular risk who may benefit from earlier or more aggressive preventive strategies. Although preliminary, our findings justify the continuation of this cohort in a longitudinal manner to assess whether baseline fetuin-A predicts future progression of atherosclerosis or deterioration of metabolic parameters. From a clinical perspective, our findings emphasize the potential value of integrating hepatic steatosis assessment and fetuin-A measurement into routine cardiovascular risk stratification for patients with type 2 diabetes mellitus. Such an approach may help identify high-risk individuals earlier and guide more personalized preventive and therapeutic strategies.

## 5. Conclusions

In conclusion, our study highlights the high prevalence of metabolic syndrome and hepatic steatosis among patients with type 2 diabetes mellitus, emphasizing their strong association with dyslipidemia, vascular changes, and cardiovascular risk. We found significant correlations between steatosis, adverse lipid profiles, and increased carotid intima–media thickness, suggesting that metabolic-associated fatty liver disease should be regarded as a systemic disorder rather than an isolated hepatic condition.

Furthermore, fetuin-A emerged as a potential biomarker linking lipid metabolism to cardiometabolic dysfunction. These findings support the importance of early detection, integrated management strategies, and closer monitoring of metabolic and vascular health in diabetic patients. Future longitudinal studies with larger cohorts are warranted to confirm these associations and to clarify the mechanistic pathways, with the perspective of developing targeted preventive and therapeutic interventions.

## Figures and Tables

**Figure 1 biomedicines-13-02098-f001:**
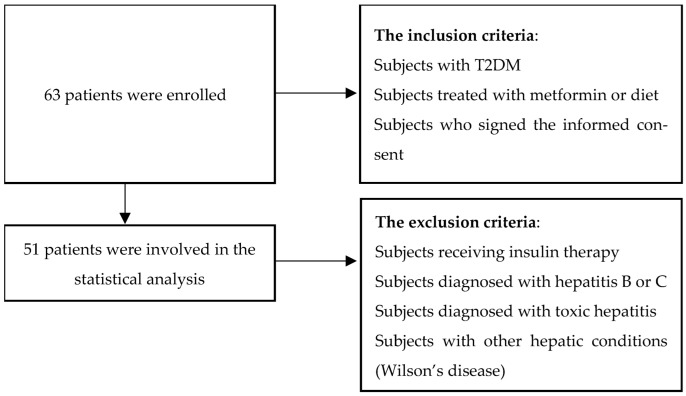
Flowchart of study design, patient inclusion, and exclusion criteria.

**Table 1 biomedicines-13-02098-t001:** International Diabetes Federation (IDF) cut-off values for metabolic syndrome diagnosis.

Parameter	IDF Cut-Off Value
Waist circumference	≥94 cm (men); ≥80 cm (women)
Triglycerides	≥150 mg/dL (1.7 mmol/L)
HDL cholesterol	<40 mg/dL (men); <50 mg/dL (women)
Blood pressure	≥130/85 mmHg
Fasting plasma glucose	≥100 mg/dL (5.6 mmol/L) or diagnosed T2DM

IDF, International Diabetes Federation.

**Table 2 biomedicines-13-02098-t002:** Baseline characteristics.

Parameters	Subjects *n* = 51	
	N	%
Female	28	54.9
Male	23	45.1
Age		
Average ± SD	62.49 ± 8.91	
Median/min–max	(62/34–86)	
≤60 years	18	35.3
>60 years	33	64.7
Atheroma plaques	4	7.8
Steatosis	30	58.8
Metabolic syndrome	40	78.4

SD = standard deviation.

**Table 3 biomedicines-13-02098-t003:** Clinical, laboratory, and imagistic investigations.

Characteristics	*n* = 51			
	Average	SD	Median	Range
BMI (kg/m^2^)	30.94	5.34	30.42	21.70–47.60
WC (cm)	103.04	11.72	100	78–133
WHR	0.95	0.07	0.96	0.80–1.08
Fetuin (ng/mL)	21.74	6.50	20.93	12.26–49.77
Glycaemia (mg/dL)	138.67	34.36	132	92–255
HbA1c	6.75	1.15	6.40	5.0–9.70
Insulinemia (mcU/mL)	13.63	9.38	10	2.0–42.60
HOMA	4.71	3.94	3.44	0.55–19.87
Creatinine (mg/dL)	0.87	0.18	0.82	0.61–1.33
Uric acid (mg/dL)	5.51	1.31	5.50	2.90–8.70
Vitamin D (ng/mL)	17.44	9.10	15.96	4.40–56.23
Total cholesterol (mg/dL)	207.37	52.82	211	110–342
LDLc (mg/dL)	136.88	46.10	136.50	54–254
HDLc (mg/dL)	43.39	11.72	40	27–75
NonHDLc (mg/dL)	163.98	55.27	164	73–310
Right CIMT (mm)	0.99	0.14	1	0.80–1.30
Left CIMT (mm)	0.99	0.18	1	0.60–1.50
A-CIMT	1.99	0.26	2	1.40–2.80

SD = standard deviation; BMI = body mass index; WC = waist circumference; WHR = waist–hip ratio; HbA1C = glycated hemoglobin; HOMA = homeostatic model assessment for insulin resistance; LDLc = low-density lipoprotein cholesterol; HDLc = high-density lipoprotein cholesterol; CIMT = carotid intima-media thickness; A-CIMT = average carotid intima-media thickness.

**Table 4 biomedicines-13-02098-t004:** Characteristics of patients with hepatic steatosis.

	Steatosis *n* = 30	Normal Liver *n* = 21	Test	*p*
Male gender	16 (53.3%)	7 (33.3%)	Chi Square Test	0.155
Age > 60 years	22 (73.3%)	11 (52.4%)	Chi Square Test	0.124
Atheroma plaques	3 (10.0%)	4 (4.8%)	Chi Square Test	0.481
Metabolic syndrome	26 (86.7%)	14 (66.7%)	Chi Square Test	0.089
BMI average ± SD median/min–max	32.48 ± 3.78 32/26.10–40.60	29.87 ± 6.04 30/21.70–47.60	One Way ANOVA test	0.087
Obesity	5 (16.7%)	5 (23.8%)	Chi Square Test	0.531
WC average ± SD median/min–max	102.53 ± 13.18 102.5/78–133	103.84 ± 9.23 104/90–120	One Way ANOVA test	0.708
High WC	25 (83.3%)	18 (94.7%)	Chi Square Test	0.820
WHR average ± SD median/min–max	0.96 ± 0.06 0.97/0.80–1.08	0.94 ± 0.07 0.94/15–30	One Way ANOVA test	0.200
WHR > 1	22 (73.3%)	12 (63.2%)	Chi Square Test	0.232
Fetuin average ± SD median/min–max	21.58 ± 7.29 21.59/12.26–49.77	21.95 ± 5.34 21.96/13.47–35.63	One Way ANOVA test	0.844

SD = standard deviation; BMI = body mass index; WC = waist circumference; WHR = waist–hip ratio.

**Table 5 biomedicines-13-02098-t005:** Characteristics of patients with hepatic steatosis.

	Steatosis (*n* = 30)	Normal Liver (*n* = 21)	Test	*p*
Glycaemia average ± SD median/min–max	137.0 ± 34.86 137/100–255	141.05 ± 34.34 141/92–234	One Way ANOVA test	0.683
High glycaemia	19 (63.3%)	13 (61.9%)	Chi Square Test	0.917
HbA1c average ± SD median/min–max	6.59 ± 1.02 6.50/5.50–9.50	6.96 ± 1.31 6.96/5.0–9.70	One Way ANOVA test	0.273
HbA1c > 6	22 (73.3%)	17 (85.0%)	Chi Square Test	0.320
Insulinemia average ± SD median/min–max	13.12 ± 9.74 13.13/2.0–42.60	14.51 ± 8.93 14.51/4.01–38.10	One Way ANOVA test	0.273
HOMA average ± SD median/min–max	4.62 ± 4.29 4.63/0.55–19.87	4.86 ± 2.99 4.87/1.07–11.96	One Way ANOVA test	0.840
Creatinin average ± SD median/min–max	0.87 ± 0.16 0.88/0.62–1.33	0.87 ± 0.20 0.86/0.61–1.29	One Way ANOVA test	0.948
Number of components for MS average ± SD median/min–max	4 ± 1 4/2–5	3 ± 1 3/1–5	One Way ANOVA test	0.003
MS	26 (86.7%)	14 (66.7%)	Chi Square Test	0.089
Uric acid average ± SD median/min–max	5.79 ± 1.22 5.79/3.20–8.70	5.12 ± 1.37 5.12/2.90–8.20	One Way ANOVA test	0.072
Uric acid > 6	14 (46.7%)	4 (19.0%)	Chi Square Test	0.038
Vit D average ± SD median/min–max	19.42 ± 10.14 19.42/4.40–56.23	14.88 ± 6.96 14.88/4.40–29.74	One Way ANOVA test	0.072
TC average ± SD median/min–max	219.13 ± 56.20 219/110–342	190.57 ± 43.49 190/126–306	One Way ANOVA test	0.050
TC > 200	20 (66.7%)	9 (42.9%)	Chi Square Test	0.091
LDLc average ± SD median/min–max	146.10 ± 50.0 146/54–254	124.14 ± 37.60 124/56–206	One Way ANOVA test	0.097
LDLc > 100	23 (76.7%)	16 (76.2%)	Chi Square Test	0.969
HDLc average ± SD median/min–max	38.30 ± 9.30 38.5/27–67	50.67 ± 11.12 51/33–75	One Way ANOVA test	0.001
HDLc < 40	22 (73.3%)	3 (14.3%)	Chi Square Test	0.001
Non-HDLc average ± SD median/min–max	180.83 ± 58.49 181/73–310	139.90 ± 40.53 140/82–233	One Way ANOVA test	0.008
Non-HDLc > 160	20 (66.7%)	7 (33.3%)	Chi Square Test	0.017
Triglycerides average ± SD median/min–max	188.83 ± 89.83 188.5/71–429	124.10 ± 42.50 124/44–201	One Way ANOVA test	0.004
Right CIMT average ± SD median/min–max	1.0 ± 0.15 1.0/0.80–1.30	0.98 ± 0.13 0.98/0.80–1.20	One Way ANOVA test	0.505
Right CIMT > 1	12 (40.0%)	6 (28.6%)	Chi Square Test	0.398
Left CIMT average ± SD median/min–max	1.03 ± 0.20 1.03/0.60–1.50	0.95 ± 0.14 0.98/0.80–1.20	One Way ANOVA test	0.048
Left CIMT > 1	15 (50.0%)	4 (19.0%)	Chi Square Test	0.021
A-CIMT average ± SD median/min–max	1.01 ± 0.15 0.99/0.70–1.40	1.02 ± 0.25 0.99/0.75–2.0	One Way ANOVA test	0.929
A-CIMT > 1	12 (40.0%)	6 (28.6%)	Chi Square Test	0.398
Right CIMT average ± SD median/min–max	1.0 ± 0.15 1.0/0.80–1.30	0.98 ± 0.13 0.98/0.80–1.20	One Way ANOVA test	0.505
Right CIMT > 1	12 (40.0%)	6 (28.6%)	Chi Square Test	0.398
Left CIMT average ± SD median/min–max	1.03 ± 0.20 1.03/0.60–1.50	0.95 ± 0.14 0.98/0.80–1.20	One Way ANOVA test	0.048
Left CIMT > 1	15 (50.0%)	4 (19.0%)	Chi Square Test	0.021
A-CIMT average ± SD median/min–max	1.01 ± 0.15 0.99/0.70–1.40	1.02 ± 0.25 0.99/0.75–2.0	One Way ANOVA test	0.929

SD = standard deviation; HbA1C = glycated hemoglobin; HOMA = homeostatic model assessment for insulin resistance; LDLc = low-density lipoprotein cholesterol; HDLc = high-density lipoprotein cholesterol; TC = total cholesterol; MS = metabolic syndrome; CIMT = carotid intima-media thickness; A-CIMT = average carotid intima-media thickness.

**Table 6 biomedicines-13-02098-t006:** Multivariate analysis of factors related to fetuin.

Model		Unstandardized Coefficients		Standardized Coefficients	t	*p*
		t	*p*	Beta		
1	Constant	21.673	3.796		5.710	0.000
	TC	0.008	0.018	−0.001	−0.005	0.996
2	Constant	26.008	4.220		6.164	0.000
	TC	−0.107	0.054	−0.868	−1.976	0.054
	LDLc	0.130	0.062	0.915	2.082	0.043
3	Constant	26.968	5.456		4.493	0.000
	TC	−0.104	0.056	−0.847	−1.880	0.066
	LDLc	0.126	0.064	0.888	1.959	0.050
	HDLc	−0.023	0.081	−0.041	−0.282	0.779
4	Constant	24.843	5.638		4.406	0.000
	TC	−0.124	0.057	−1.003	−2.173	0.035
	LDLc	0.138	0.064	0.969	2.135	0.038
	HDLc	0.014	0.085	0.025	0.163	0.871
	Triglycerides	0.019	0.014	0.216	1.339	0.187
5	Constant	19.243	7.081		2.717	0.009
	TC	−0.130	0.057	−1.050	−2.285	0.027
	LDLc	0.146	0.064	1.031	2.276	0.028
	HDLc	0.068	0.094	0.121	0.720	0.475
	Triglycerides	0.009	0.016	0.105	0.578	0.566
	Components for MS	1.345	1.042	0.243	1.291	0.204

LDLc = low-density lipoprotein cholesterol; HDLc = high-density lipoprotein cholesterol; TC = total cholesterol; MS = metabolic syndrome.

## Data Availability

Data will be available by contacting the corresponding author. Data is not publicly available due to privacy.

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
