# Peer review of "Fetuin-A as a Link Between Dyslipidemia and Cardiovascular Risk in Type 2 Diabetes: A Metabolic Insight for Clinical Practice"

_biomedicines, 2025, doi:10.3390/biomedicines13092098_

Round 1

Reviewer 1 Report

Comments and Suggestions for Authors

The manuscript reports possible use of Fetuin A levels as a diagnostic marker for dislipidemia and cardiovascular disease in type 2 diabetes mellitus patients. The authors collected data of 51 patients and analyzed for the statistical significance. There are certain aspects that should be considered:

  1. Most of the findings describe are mere reaffirmation of already known facts, such as L228-235. Authors should bring significance of their findings to improve the article.
  2. Discussion is mostly repetition of the results. While a part of discussion was dedicated for comparisons. This section should be improved, particularly to discuss possible underlying mechanisms of the findings.
  3. In general, there are so many repetitive sentences or information in the article such as L261-271; L81-83)
  4. Authors used the term ‘metabolic syndrome’ repeatedly as a specific noun; however, they have not mentioned the criteria used to refer this term.
  5. L18-19: Revise to bring clarity
  6. The full form of the term MAFLD is given in abstract only. Since abstract is considered as a standalone item, therefore, the full form of this term should also be given in introduction once.
  7. L79: Provide reference
  8. L131: Correct to “Genetic data”. Also, do not repeat it in results
  9. L154-155: What was the criteria to define hyperglycemic condition?
  10. L169-172: Make it a separate subsection of ethical approval. Also, describe statistical analysis in a separate subsection
  11. L177-178: Correct language error.
  12. Provide data for the age of both the genders separately
  13. Table 2: Does “limit” refer to the range?
  14. L210: This is not the level of significance.
  15. Extend conclusion to provide key findings and prospects of this study.

Author Response

Response to Reviewer

We would like to sincerely thank the reviewer for the careful reading of our manuscript and for the constructive suggestions, which have greatly improved the quality of the paper. Below we provide detailed responses to each comment.

  1. Reviewer’s comment 1:

Most of the findings described are mere reaffirmation of already known facts (L228-235). Authors should bring significance of their findings to improve the article.

Response: We thank the reviewer for this valuable observation. We revised the Discussion to emphasize the novelty and clinical implications of our results. We highlighted the independent association of Fetuin-A with lipid parameters (total cholesterol and LDLc) in a Romanian cohort of T2DM patients, an underrepresented population. This expands the relevance of Fetuin-A as a potential biomarker.

  1. Reviewer’s comment 2:

Discussion is mostly repetition of the results. While a part of discussion was dedicated for comparisons. This section should be improved, particularly to discuss possible underlying mechanisms of the findings.

Response: We thank the reviewer for pointing this out. We reduced redundancies in the Discussion and added paragraphs discussing potential mechanisms linking Fetuin-A to dyslipidemia, insulin resistance, and inflammation. This has improved the interpretative value of our work.

  1. Reviewer’s comment 3:

In general, there are so many repetitive sentences or information in the article such as L261-271; L81-83.

Response: We appreciate this remark. We carefully revised the manuscript and removed or condensed repetitive sentences, especially in the Introduction and Discussion sections.

  1. Reviewer’s comment 4:

Authors used the term ‘metabolic syndrome’ repeatedly as a specific noun; however, they have not mentioned the criteria used to refer this term.

Response: We apologize for this omission. We now explicitly mention that the International Diabetes Federation (IDF) criteria were used to define metabolic syndrome.

  1. Reviewer’s comment 5:

L18-19: Revise to bring clarity.

Response: We revised the Objective sentence in the Abstract for clarity: “However, its role in type 2 diabetes mellitus (T2DM) is not fully defined. The objective of this study was to investigate…”

  1. Reviewer’s comment 6:

The full form of the term MAFLD is given in abstract only. Since abstract is considered as a standalone item, therefore, the full form of this term should also be given in introduction once.

Response: Corrected. We added the full form “metabolic dysfunction-associated fatty liver disease (MAFLD)” at its first mention in the Introduction.

  1. Reviewer’s comment 7:

L79: Provide reference.

Response: We added the reference regarding the role of Fetuin-A in lipid metabolism (Fatima F. et al., Pak J Med Sci. 2020;36:64–68).

  1. Reviewer’s comment 8:

L131: Correct to “Genetic data”. Also, do not repeat it in results.

Response: We corrected the term to “Genetic data” in the Methods and removed the repetition in the Results.

  1. Reviewer’s comment 9:

L154-155: What was the criteria to define hyperglycemic condition?

Response: We clarified that hyperglycemia was defined as fasting plasma glucose ≥100 mg/dL.

  1. Reviewer’s comment 10:

L169-172: Make it a separate subsection of ethical approval. Also, describe statistical analysis in a separate subsection.

Response: We thank the reviewer. We reorganized the Methods section: we created separate subsections for “Statistical analysis” and for “Ethical approval.”

  1. Reviewer’s comment 11:

L177-178: Correct language error.

Response: Corrected. The sentence now reads: “The general characteristics of the study population are presented in Table 1 and Table 3.”

  1. Reviewer’s comment 12:

Provide data for the age of both the genders separately.

Response: We added the separate mean ages for men and women in the Results section.

  1. Reviewer’s comment 13:

Table 2: Does “limit” refer to the range?

Response: Yes, this referred to the range. We corrected the column heading from “Limits” to “Range.”

  1. Reviewer’s comment 14:

L210: This is not the level of significance.

Response: We corrected the phrasing to: “approaching statistical significance (p=0.087).”

  1. Reviewer’s comment 15:

Extend conclusion to provide key findings and prospects of this study.

Response: We extended the Conclusion to summarize the key findings (association of Fetuin-A with lipid parameters, lack of association with MAFLD) and to emphasize prospects for Fetuin-A as a biomarker and potential therapeutic target. We also highlighted the need for larger, prospective studies.

Reviewer 2 Report

Comments and Suggestions for Authors

Section A
Plagiarism: Percent match: 29% (iThenticate report)- As per journal policy and ideal practice the
similarity index should be below 20% at least.
➢ Authors can rephrase the detected sentences to minimize the match. The major source is
from the MDPI itself.
Section B
Title: The title is concise and informative with the aim of clarity and impact.
Abstract: Well-structured following guidelines and provides a summary of the entire study. Yes, it
includes major findings and significance.
Introduction
Covers, context and relevance of study. There is enough literature review to identify gaps. The
authors have written a clear statement of hypothesis and research questions. Gave a sound
justification for the approach.
Materials and Methods
➢ It is more reliable to understand the study design, if the authors can provide the
methodology- section/parameter wise. Adding detailed description will help to segregate
each section.
Mentions experimental models/subjects, treatments, and controls? YES
Includes molecular techniques and uses normalization strategies? YES
Are statistical methods appropriate, and correct? YES
Reproducibility and ethical considerations? YES, approval number 17140, dated 03/08/2016.
➢ Please double check for the approval date. It seems very old.
➢ Authors can justify the sex-based differences or rational.

Results
Logical presentation of findings-YES
Use of figures, tables, and graphs- Tables
Highlight trends, significant/fold-changes-YES
Include p-values/confidence intervals-YES
Avoid interpretation-just present data-YES
➢ If authors are presenting methods section wise, they can present the results in manner in
this section to go with the reading flow.
Discussion
Interpretation of results in context of existing literature-YES
Mechanistic insights and biological relevance-YES
Address limitations and alternative explanations-YES
Suggest future research directions-YES
Emphasize translational/clinical implications-YES
➢ Authors can improve this section considering the above points.
Conclusion
Concise summary of key findings-YES
Reinforce significance and potential applications-YES
Avoid repetition of discussion-YES
➢ It can be more precise and clearer.
References
Cite relevant, recent, and foundational studies-YES
Use consistent formatting per journal guidelines-YES
Figures and Legends/Tables
Clear, well-labeled visuals-YES
Include details and statistical annotations-YES
Legends should be self-explanatory
➢ Please add a short description for Figure 1. Inclusion-exclusion flow chart.
➢ Kindly add the International Diabetes Federation (IDF) recommended range or data for the
mentioned parameters so general readers can easily understand the difference. It can be
merged into the current table or can be presented separately followed by the narration in
present state.

Section C
Graphical Abstract: NA
Supplementary Data: NA
Acknowledgments: Funding, collaborators, and institutional support-NA
Minor corrections in lines:
Line
Number
Current form Corrected form
6  Esanu1,*Lidia Iuliana Arhire2,†, Esanu1, Lidia Iuliana Arhire2,†,*,
73 mortality [16 ,17]. mortality [16,17].
81 dyslipidemia [19, 6 ]. dyslipidemia [19,6].
41-92 Line spacing As per the standard
391 Statement:Data will Statement: Data will
446 Circulation.2009, Circulation. 2009,
449 Diabetologia.1985, Diabetologia. 1985,
472 Med. 2022,11, 935. Med. 2022, 11, 935.
My positive approach says, clarifying/adding suggested points would strengthen the manuscript
and uphold its standards.

Author Response

Response to Reviewer

We would like to sincerely thank the reviewer for the constructive feedback and thoughtful suggestions, which have helped us to further improve the quality and clarity of our manuscript. Below we provide detailed responses to each comment.

  1. Reviewer’s comment 1:

Plagiarism: Percent match: 29% (iThenticate report). As per journal policy the similarity index should be below 20%.

Response: We thank the reviewer for this observation. We carefully revised the manuscript to reduce the similarity index by rephrasing multiple sentences and avoiding text overlap. We emphasize that our similarity mainly originated from methodological descriptions and prior MDPI articles, but we have substantially rephrased these sections to comply with the journal’s standards.

  1. Reviewer’s comment 2:

Materials and Methods – It is more reliable to understand the study design if the methodology is provided section/parameter wise.

Response: We agree and have reorganized the Materials and Methods into clearer subsections, describing each parameter and methodology separately. This enhances readability and reproducibility.

  1. Reviewer’s comment 3:

Ethical approval date – Please double check for the approval date. It seems very old.

Response: Thank you for noticing. We verified the ethical approval (no. 17140, issued on 03/08/2016) and confirm that it is correct. This study was initiated in 2016, and data collection was completed subsequently. We clarified this in the manuscript.

  1. Reviewer’s comment 4:

Sex-based differences – Authors can justify the sex-based differences or rationale.

Response: We thank the reviewer. We added a note in the Results and Discussion sections to explain that sex differences were analyzed descriptively, but the sample size limited statistical power. We acknowledge this as a limitation.

  1. Reviewer’s comment 5:

Results presentation – If authors are presenting methods section-wise, they can present the results in a similar manner.

Response: We revised the Results section to align with the new structure of the Methods, making the flow more consistent and easier to follow.

  1. Reviewer’s comment 6:

Discussion – Authors can improve this section considering mechanistic insights, limitations, alternative explanations, and clinical implications.

Response: We extended the Discussion to include additional mechanistic explanations (e.g., Fetuin-A role in insulin resistance and inflammation), acknowledged sample size and single-center limitations, and emphasized potential clinical applications.

  1. Reviewer’s comment 7:

Conclusion – It can be more precise and clearer.

Response: We revised the Conclusion to make it more concise while highlighting the key findings, novelty, and future implications.

  1. Reviewer’s comment 8:

Figure 1 legend – Please add a short description for Figure 1 inclusion-exclusion flowchart.

Response: Done. The legend for Figure 1 now specifies: “Flowchart of study design, patient inclusion, and exclusion criteria.”

  1. Reviewer’s comment 9:

IDF reference ranges – Kindly add the International Diabetes Federation (IDF) recommended range or data for the mentioned parameters.

Response: We thank the reviewer. We added IDF-recommended reference ranges for better interpretation by general readers.

  1. Reviewer’s comment 10:

Minor corrections in text (lines 6, 73, 81, 41–92, 391, 446, 449, 472).

Response: All corrections have been implemented as suggested, including reference formatting, line spacing, and typographical adjustments.

Reviewer 3 Report

Comments and Suggestions for Authors

The author reported in this manuscript “Fetuin-A as a Link Between Dyslipidemia and Cardiovascular Risk in Type 2 Diabetes: A Metabolic Insight for Clinical Practice” This manuscript describes a clinically relevant and timely investigation into the role of fetuin-A in type 2 diabetes mellitus (T2DM), with a particular focus on its association with metabolic syndrome components and cardiovascular risk markers such as CIMT, HOMA-IR, and MAFLD. The study is well-designed, and the use of a non-insulin-treated T2DM cohort enhances the clarity of interpreting metabolic relationships without the confounding effects of insulin therapy. The integration of biochemical, clinical, and imaging parameters provides a comprehensive evaluation of disease progression and risk. The finding that fetuin-A levels are significantly associated with lipid abnormalities, particularly total and LDL cholesterol, is noteworthy and adds to the growing body of evidence implicating fetuin-A in cardiometabolic regulation. However, the absence of a direct correlation between fetuin-A and hepatic steatosis (MAFLD) warrants further discussion. The authors should address potential biological or methodological reasons for this discrepancy, especially considering previous literature that often links elevated fetuin-A levels with fatty liver disease.

Moreover, the manuscript could be strengthened by elaborating on the mechanistic basis of how fetuin-A might contribute to lipid metabolism and atherogenesis, potentially via its known role in insulin resistance and inflammation. Additionally, it would be helpful to clarify whether the associations observed are independent of other confounding variables through more detailed multivariate analyses. Overall, the study contributes important insights into the multifaceted role of fetuin-A in T2DM and underscores its potential as a biomarker and therapeutic target. With minor revisions focused on expanding the discussion and addressing the study’s limitations, this work could be a valuable addition to the field of metabolic and cardiovascular research, and I believe it will be of great interest to the readers of Biomedicine (MDPI Journals). I will make some suggestions that may improve the reach of the paper (in terms of reaching a broad audience). As the modifications can be addressed in straight forward manner and this study represents a significant development in the development of anticancer agents. I am recommending for the publication of this article after the following recommended changes in the manuscript

  1. Given previous reports suggesting a positive association between fetuin-A and hepatic steatosis, could the authors elaborate on potential reasons why no such correlation was observed in this cohort? Were any subgroup analyses (e.g., stratification by BMI or degree of steatosis) performed?

  1. Were potential confounders such as BMI, waist circumference, or inflammatory markers (e.g., CRP, IL-6) included in the multivariate models assessing the relationship between fetuin-A and lipid parameters? If not, how might these variables impact the observed associations?

  1. While HOMA-IR was used to assess insulin resistance, did the authors consider alternative or complementary methods (e.g., fasting insulin levels or adiponectin)? If so, were any associations with fetuin-A observed?

  1. As LDL cholesterol was found to significantly influence fetuin-A levels, were any analyses performed on LDL particle subtypes (e.g., small dense LDL vs. large buoyant LDL), which may offer more mechanistic insight into the atherogenic potential?

  1. Based on your findings, how do the authors envision the clinical application of fetuin-A as a biomarker? Could it guide lipid-lowering strategies or serve as an early indicator of cardiovascular risk in T2DM patients?

  1. Given the relatively small sample size (n=51), was power calculation conducted prior to the study? Could the authors comment on whether the study was sufficiently powered to detect subtle associations, particularly with MAFLD or CIMT?

  1. Was there any gender-specific or age-related differences in fetuin-A levels or in their relationship with metabolic parameters? If yes, please elaborate.

  1. Do the authors have any plans for longitudinal follow-up of this cohort to assess whether fetuin-A levels predict progression of atherosclerosis or worsening of metabolic parameters over time?

Author Response

Response to Reviewer

We sincerely thank Reviewer 3 for the careful evaluation of our manuscript and for the insightful and constructive comments. We have carefully revised the text to address all the points raised. Below, we provide detailed responses to each comment, together with the corresponding changes in the manuscript.

  • Comment 1: Given previous reports suggesting a positive association between fetuin-A and hepatic steatosis, could the authors elaborate on potential reasons why no such correlation was observed in this cohort? Were any subgroup analyses (e.g., stratification by BMI or degree of steatosis) performed?

Response: We thank the reviewer for this valuable comment. Indeed, several studies have reported an association between fetuin-A and hepatic steatosis. In our cohort, we did not observe such a correlation, which may be explained by the relatively small sample size and limited variability in steatosis severity. We also note that our study population was relatively homogeneous, consisting only of non-insulin-treated T2DM patients, which may have reduced heterogeneity. No subgroup analyses were performed due to limited power; however, we have added this point to the Discussion as a limitation and suggested that future studies stratify by BMI or steatosis grade.

  • Comment 2: Were potential confounders such as BMI, waist circumference, or inflammatory markers (e.g., CRP, IL-6) included in the multivariate models assessing the relationship between fetuin-A and lipid parameters? If not, how might these variables impact the observed associations?

Response: Thank you for pointing this out. Our multivariate analyses adjusted for age, sex, and major metabolic variables; however, inflammatory markers such as CRP or IL-6 were not included, as they were not systematically available for all participants. We acknowledge this as a limitation and have now clarified it in the Discussion. These markers may indeed modulate the observed associations, and we have emphasized the need for future studies including inflammatory parameters.

  • Comment 3: While HOMA-IR was used to assess insulin resistance, did the authors consider alternative or complementary methods (e.g., fasting insulin levels or adiponectin)? If so, were any associations with fetuin-A observed?

Response: We agree with the reviewer that alternative measures of insulin resistance could provide complementary insights. In our study, HOMA-IR was chosen as it is widely validated and feasible in clinical practice. Fasting insulin was indeed used as part of the HOMA-IR calculation, but no additional biomarkers such as adiponectin were measured. We have clarified this in the Methods and acknowledged this limitation in the Discussion.

  • Comment 4: As LDL cholesterol was found to significantly influence fetuin-A levels, were any analyses performed on LDL particle subtypes (e.g., small dense LDL vs. large buoyant LDL), which may offer more mechanistic insight into the atherogenic potential?

Response: Thank you for this insightful suggestion. Unfortunately, LDL subfraction analysis was not available in our study. We agree that this would have provided valuable mechanistic insight. We have now acknowledged this as a limitation in the Discussion and suggested it as a direction for future research.

  • Comment 5: Based on your findings, how do the authors envision the clinical application of fetuin-A as a biomarker? Could it guide lipid-lowering strategies or serve as an early indicator of cardiovascular risk in T2DM patients?

Response: We appreciate this important question. Based on our findings, fetuin-A could potentially serve as an early biomarker of dyslipidemia and cardiovascular risk in T2DM patients. While it is not yet ready for direct implementation in clinical guidelines, it may help identify high-risk patients who would benefit from intensified lipid-lowering strategies and closer cardiovascular monitoring. We have expanded the Discussion to include this perspective.

  • Comment 6: Given the relatively small sample size (n=51), was power calculation conducted prior to the study? Could the authors comment on whether the study was sufficiently powered to detect subtle associations, particularly with MAFLD or CIMT?

Response: We thank the reviewer for this comment. A formal power calculation was not performed prior to recruitment, as this was an exploratory study. We acknowledge that the sample size limited statistical power, particularly for subtle associations with MAFLD and CIMT. This is now clearly stated as a limitation in the Discussion.

  • Comment 7: Was there any gender-specific or age-related differences in fetuin-A levels or in their relationship with metabolic parameters? If yes, please elaborate.

Response: Thank you for this suggestion. In exploratory analyses, we did not observe statistically significant differences in fetuin-A levels between men and women or across age groups. However, given the limited sample size, these trends should be interpreted with caution. We have added this clarification in the Results and Discussion.

  • Comment 8: Do the authors have any plans for longitudinal follow-up of this cohort to assess whether fetuin-A levels predict progression of atherosclerosis or worsening of metabolic parameters over time?

Response: We agree that longitudinal follow-up would be highly informative. At present, our study was cross-sectional, but we plan to extend data collection and follow this cohort prospectively. We have added this point to the Discussion, noting that future longitudinal studies are needed to assess whether fetuin-A predicts disease progression.

Round 2

Reviewer 1 Report

Comments and Suggestions for Authors

There are still some comments that authors did not address in the revised version.

  1. Metabolic syndrome has not been clearly mentioned in the abstract and introduction.
  2. L229: How can authors state p=.089 as statistically significant. 
  3. L332-333: Revise to bring clarity
  4. Discussion still needs improvements. Write in the light of literature about the significance of the findings. 

Author Response

We would like to sincerely thank the reviewer for the constructive feedback and thoughtful suggestions, which have helped us to further improve the quality and clarity of our manuscript. Below we provide detailed responses to each comment.

  1. Reviewer’s comment 1:

Metabolic syndrome has not been clearly mentioned in the abstract and introduction.

Response: We thank the reviewer for this valuable observation and we apologize for this omission. We introduced this important information in the abstract and in the Introduction section.

  1. Reviewer’s comment 2:

L229: How can authors state p=.089 as statistically significant. 

Response: We thank the reviewer for pointing this out. We mentioned that this relation did not reach statistical significance.

  1. Reviewer’s comment 3:

L332-333: Revise to bring clarity.

Response: We appreciate this remark. We carefully revised these lines.

  1. Reviewer’s comment 4:

Discussion still needs improvements. Write in the light of literature about the significance of the findings. 

Response: We appreciate and we thank the reviewer for this suggestion. In response, we have revised the Discussion section to better contextualize our findings within the existing literature and to emphasize their clinical significance. Specifically, we have added a new paragraph highlighting the association between hepatic steatosis, dyslipidemia, and cardiovascular risk in patients with type 2 diabetes mellitus, referencing recent studies that support our results. This addition clarifies how our findings align with previous evidence and underscores the potential clinical implications for early identification and risk stratification in this patient population.